# Ultrahigh dynamic range and low noise figure programmable integrated microwave photonic filter

Okky Daulay[1,5], Gaojian Liu [1,2,5], Kaixuan Ye[1], Roel Botter [1], Yvan Klaver[1], Qinggui Tan[2], Hongxi Yu[2], Marcel Hoekman[3], Edwin Klein[3], Chris Roeloffzen [3], Yang Liu [4] & David Marpaung [1] ✉

Microwave photonics has adopted a number of important concepts and technologies over the recent pasts, including photonic integration, versatile programmability, and techniques for enhancing key radio frequency performance metrics such as the noise figure and the dynamic range. However, to date, these aspects have not been achieved simultaneously in a single circuit. Here, we report a multi-functional photonic integrated circuit that enables programmable filtering functions with record-high performance. We demonstrate reconfigurable filter functions with record-low noise figure and a RF notch filter with ultra-high dynamic range. We achieve this unique feature using versatile complex spectrum tailoring enabled by an all integrated modulation transformer and a double injection ring resonator as a multi-function optical filtering component. Our work breaks the conventional and fragmented approach of integration, functionality and performance that currently prevents the adoption of integrated MWP systems in real applications.

As radio frequency (RF) and microwave systems are moving forward into cognitive operation, reconfigurable filter will become a key component to enable the full potential of these systems' performance[1–3]. This filter can intelligently operate to differentiate the RF signal of interest from the interferers[4–6]. There is a need of developing reconfigurable filter for modern RF systems to address the challenges impeding the use of active electronically scanned array (AESA) that operate at a wide range of frequencies in dense RF environment[7]. The filter is aimed to make the array in AESA more resistant to interference before signal processing[8]. Integrated microwave photonic (MWP) can offer significant advantages to realize advanced concepts of reconfigurable filter for multi-band, all spectrum communications[9] and broadband programmable front-ends[10], which are important for modern RF communications (i.e. cognitive radio). To play a key role in modern RF applications, integrated MWP circuits need to simultaneously show advanced programmability and exceptional performance in terms of low losses, low noise figure, and high

dynamic range in a reduced footprint[11–15]. In recent pasts, a number of programmable integrated MWP filters have widely been demonstrated[16–20]. Typically, these filters were achieved in application-specific circuits, and the measured RF metrics were only sparsely reported. The values of the RF gain, noise figure (NF), and spurious-free dynamic range (SFDR) in these circuits are usually far-off from the requirements for practical RF systems.

Recently, new ways of building a programmable integrated MWP circuit have been extensively explored, for example, through mesh circuit that can synthesize a large number of functionalities through programming[21–24]. While versatile, the typical functional performance of these circuits, such as filter extinction, or passband quality and the RF metrics were significantly lower than that of application-specific circuits[25–28]. Thus, at present, the unique and well-sought combination of high integration density, versatile programmability and high RF performance remains elusive. A versatile modulation transformer (MT) that can tailor the phases and amplitudes of optical carrier and RF

[1]Nonlinear Nanophotonics Group, MESA+ Institute of Nanotechnology, University of Twente, Enschede, Netherlands. [2]China Academy of Space Technology (Xi'an), Xi'an, China. [3]LioniX International BV, Enschede, Netherlands. [4]Institute of Physics, Swiss Federal Institute of Technology Lausanne (EPFL), CH-1015 Lausanne, Switzerland. [5]These authors contributed equally: Okky Daulay, Gaojian Liu. ✉e-mail: david.marpaung@utwente.nl

sidebands has recently been identified as a key element for enhancement of dynamic range[29–32]. On the other hand, a programmable resonator such as a double-injected ring resonator (DI-RR)[33] can provide a large number of filtering functions. However, there remains a bottleneck in co-integrating components that support NF reduction and linearization with components that provide functionalities.

In this work, we demonstrate a programmable integrated MWP circuit with a unique combination of a versatile MT device and a DI-RR, realized in a low-loss silicon nitride platform. With this circuit, we show an array of RF filters in three different scenarios simultaneously with record-low NF for RF notch and RF bandpass filter, achieved using low-biasing technique[25,26] in an intensity modulator (IM)-based system, and ultra-high dynamic range for RF notch filter, achieved using on-chip linearization in a phase modulator (PM)-based system. The capability of the proposed reconfigurable filter is expected to improve the system performance across S-band through Ku-band (i.e. 2 GHz to 18 GHz) frequency range and play a key role for the realization of practical programmable integrated MWP circuit that can operate in congested RF environment.

## Results

### Programmable microwave photonics

The concept of ultrahigh dynamic range and low NF programmable integrated MWP filter explored here is illustrated in Fig. 1. The MT shapes the input modulated signal from an optical modulator to the desired modulation format that is suited for the targeted filtering function. The spectral shaping can equally provide a basis for creating bespoke phase and amplitude distribution of the optical carrier and sidebands, leading to linearization and NF reduction. The DI-RR is used as a reconfigurable spectral filtering element that can provide notch or bandpass filtering. In this way, high dynamic range and programmable filtering can be achieved simultaneously.

The MT is implemented as an asymmetric Mach-Zehnder interferometer (aMZI) loaded with 3 ring resonators as a spectral de-interleaver[29,34], combined with a tunable attenuator and a phase shifter (see Supplementary Information A). The DI-RR is an add-drop ring resonator that is double-injected from single input, hence capable of synthesizing a large number of responses (see Supplementary information B). The schematic of the entire circuit overlaid on a photograph

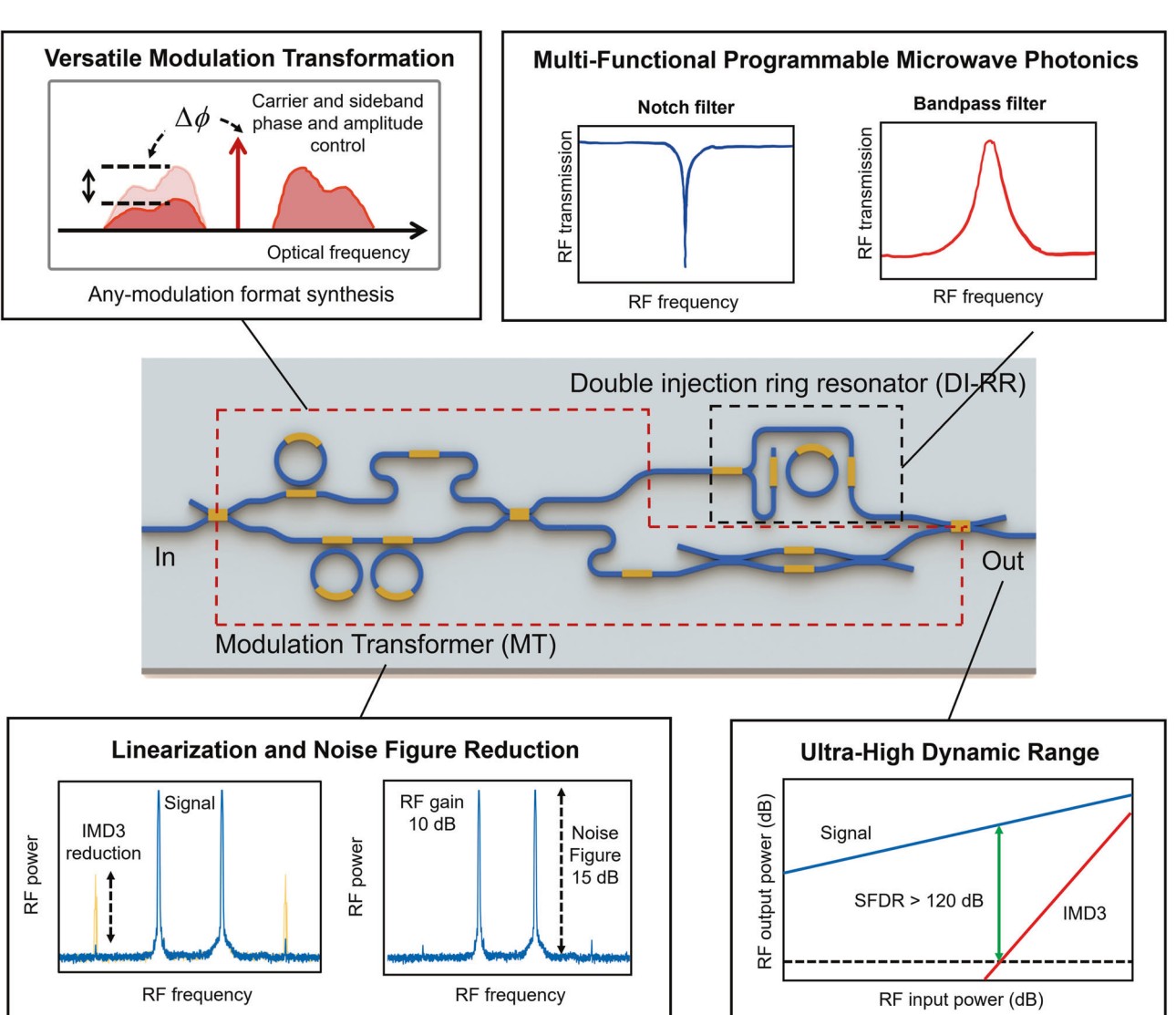

**Fig. 1 | Artistic impression of an ultra-high dynamic range programmable integrated MWP circuit.** The circuit contains of a versatile modulation transformer (MT) to independently tailor the phase and amplitude of optical modulation spectrum and an equally versatile double-injection ring resonator (DI-RR) to synthesize a variety of responses, including programmable RF filters. The combination of MT and DI-RR also allows for linearization through cancellation of intermodulation distortion (IMD) and noise figure (NF) reduction through low biasing and carrier suppression technique, leading to ultra-high dynamic range. SFDR: spurious-free dynamic range.

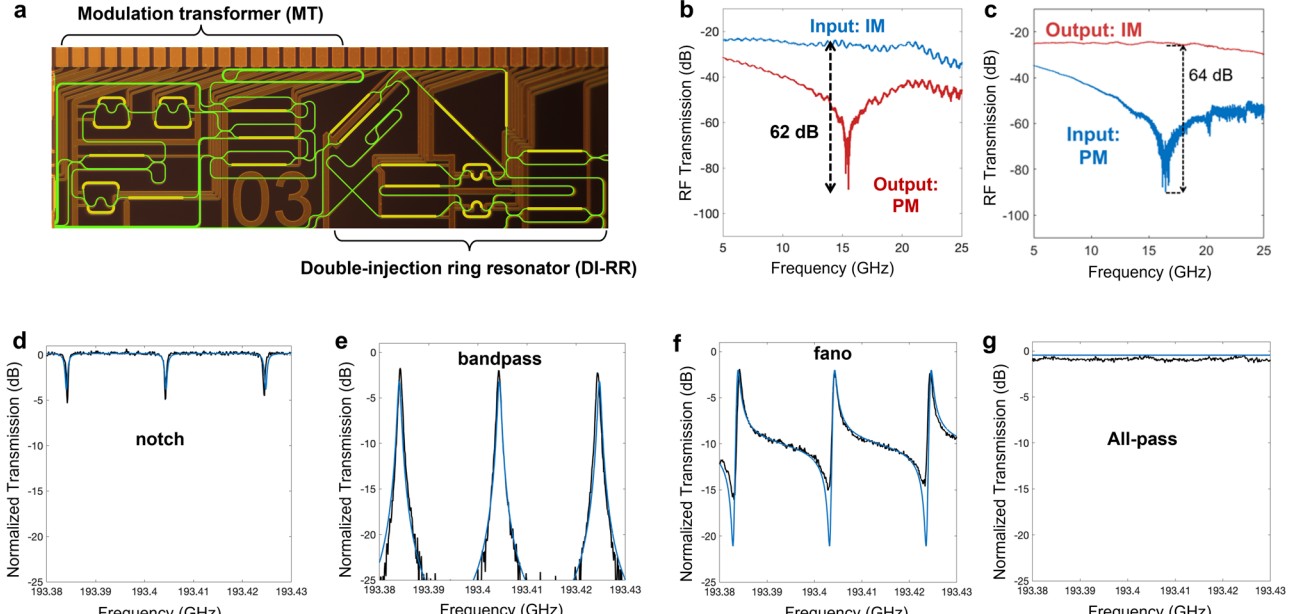

**Fig. 2 | Programmable integrated MWP circuit. a** The optical image of the programmable photonic chip with outlined waveguide leads, containing of a modulation transformer (MT) and a double-injection ring resonator (DI-RR). The MT consists of a spectral de-interleaver used to isolate one RF sideband from the entire modulated spectrum. The phase shifter and tunable coupler are used to control the phase and amplitude of the isolated sideband. Modulation transformation is achieved after recombination at the output. (see Suppplementary Information A for details of the MT.) **b** Experimental result of intensity-to-phase modulation (IM-PM) conversion with 62 dB extinction, achieved using the MT with IM input. **c** Measured phase-to-intensity modulation (PM-IM) conversion with 64 dB extinction, achieved using the MT with PM input. The DI-RR circuit is used to synthesize multiple filter functionalities. **d–g** Simulated (blue line) and measured (black line) selected responses of the silicon nitride DI-RR when tuned to exhibit notch filter (**d**), bandpass filter (**e**), Fano-like response (**f**), and all-pass response (**g**) (see Supplementary Information B for details of the DI-RR).

of the fabricated silicon nitride chip is shown in Fig. 2a. The waveguide propagation loss in the circuit is 0.1 dB/cm and the fiber-to-chip coupling loss is 1.1 dB/facet.

We demonstrate the programmability of our circuit by showing the conversion of modulation formats, as well as the flexible switching among various filtering functions. We exemplify the intensity modulation-to-phase modulation (IM-PM) conversion by inverting the phase of one sideband by $\pi$, which leads to sideband cancellation with 62 dB extinction in direct photodetection (Fig. 2b). Conversely, with a phase modulator input, the circuit can achieve PM-IM conversion with an increase in the extinction of up to 64 dB (Fig. 2c). The DI-RR, on the other hand, can synthesize a variety of responses from a single output port, including notch filter, bandpass filter, Fano-like response, and an all-pass response[35], as shown in Fig. 2d–g. Benefiting from the low propagation loss of the silicon nitride waveguide, a fine spectral resolution of around 400 MHz in the filtering function can be achieved with 20 GHz free spectral range (FSR).

### High gain and low noise figure MWP filter

We carried out a series of MWP filtering experiments with a setup diagrammed in Fig. 3a. A high power optical source is sent to a low-biased IM. We then configure our MT to shape the IM spectrum into a dedicated output spectrum suitable for the filtering tasks provided by the DI-RR. In particular, we synthesized two RF filters, namely RF notch filter and RF bandpass filter with simultaneously high RF performance metrics (see Supplementary Information C for details of the extended experimental scenarios).

Figure 3b shows the measured RF notch filter with 58 dB rejection and 3-dB bandwidth of 400 MHz. The high rejection in the filter was achieved through phase cancellation[16,36] enabled by adjusting the phase and amplitude of the isolated sideband in intensity modulation-to-asymmetric dual sidebands modulation (IM-aDSB) conversion. Through low biasing of the IM, we then achieved an optimum

measured RF gain of 10 dB and a lowest measured NF of 15 dB (Fig. 3c). These high gain and low noise figure translated into a high SFDR of 116 dB Hz$^{2/3}$ when we performed a two-tone test at the frequency of 1 GHz (Fig. 3d). It can be seen that the third-order input intercept ponts (IIP3) can be estimated to be +16 dBm with the measured noise power spectral density (PSD) of −149 dBm/Hz. These results represent the best RF gain, noise figure, and SFDR combination for an MWP notch filter. The RF notch filter central frequency can be tuned from 5 to 20 GHz (Fig. 3b) with a maximum RF gain of more than 0 dB over the frequency range (see Supplementary Information D).

We then program the MT and the DI-RR to exhibit RF bandpass filter with 20 dB rejection, with up to 15 GHz tuning range (from 5 GHz to 20 GHz) (Fig. 3e), limited by the roll-off and the dispersion of the spectral de-interleaver, notably at the transition band (see Supplementary Information A). This tuning range can be feasibly increased using an improved design of (de)-interleaver with faster roll-off and a larger FSR. We measured RF gain of 1.2 dB and NF of 21.8 dB (Fig. 3f). Taking into account the measured noise PSD of −151 dBm/Hz, the SFDR at 7 GHz is 112 dB Hz$^{2/3}$ and the IIP3 can be estimated to be +16 dBm (Fig. 3g). The versatility of the MT allows for the combination of low biasing and single-sideband (SSB)-based RF bandpass filtering, which would have not been possible only with an IM or a dual-parallel Mach–Zehnder modulator (DPMZM). As a result, we can demonstrate the best combination of measured NF and SFDR for an RF bandpass filter. Importantly, these results were achieved without using any electrical amplification in the system.

### On-chip linearization for ultrahigh dynamic range

Applications of MWP systems usually require ultra-high SFDR of beyond 120 dB Hz$^{2/3}$[15]. Although such a metric can be achieved through linearization of the modulator transfer function[37–40], it remains challenging to achieve linearization simultaneously with functionalities. Here, we demonstrate simultaneous RF notch filter and linearization

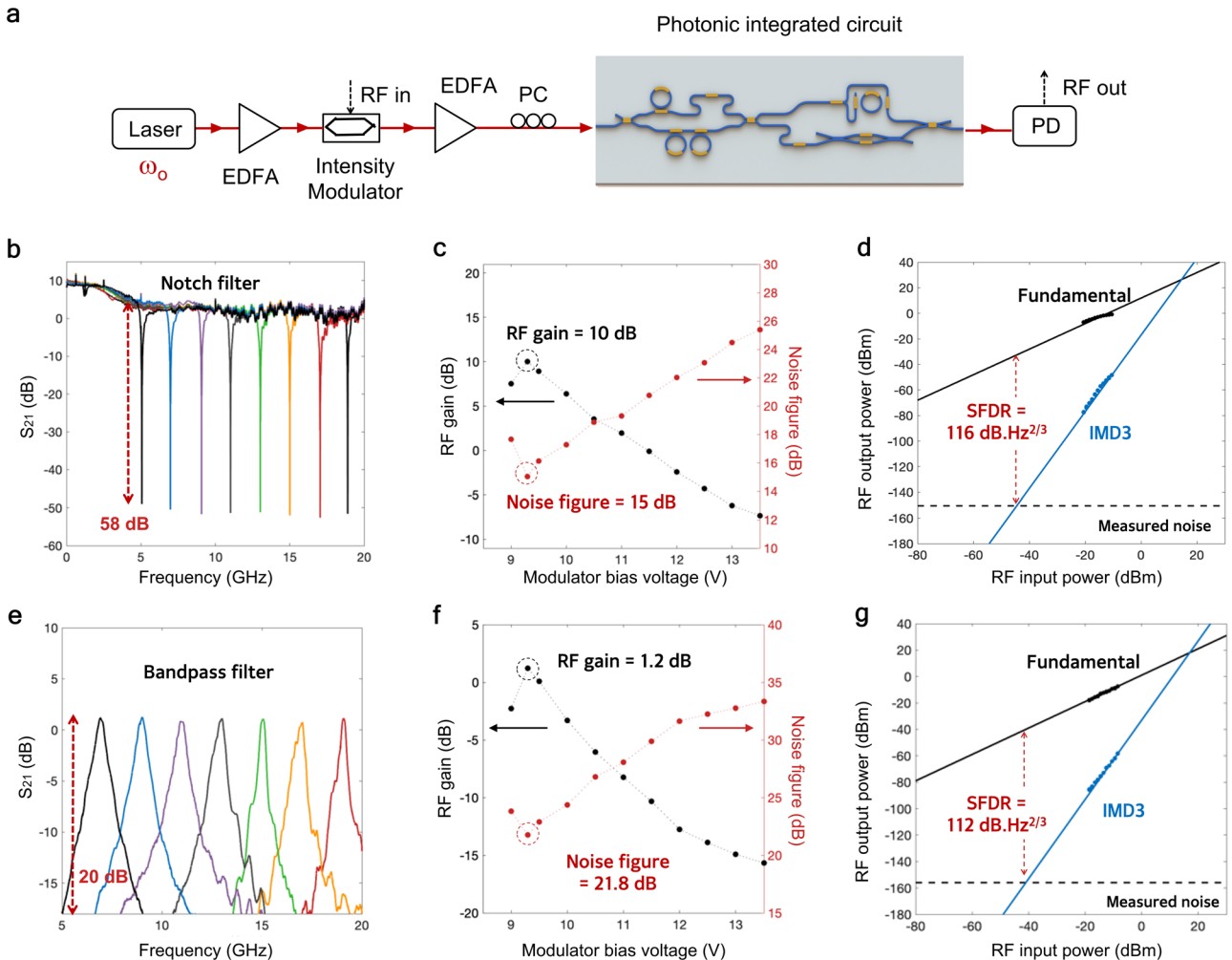

**Fig. 3 | Noise figure reduction of the programmable MWP filters. a** Experiment setup of programmable microwave photonics with enhanced performance using low-biasing Mach–Zehnder modulator (MZM). **b** High-rejection RF notch filter with 58 dB rejection. **c** The measured RF gain and noise figure (NF) of the RF notch filter. Maximum RF gain of 10 dB and a minimum NF of 15 dB are achieved through the low-biasing of the MZM. **d** The measured spurious-free dynamic range (SFDR) of the RF notch filter at 1 GHz, reaching 116 dB Hz$^{2/3}$. **e** Single-sideband (SSB) RF bandpass filter with 20 dB rejection, obtain using optical carrier re-insertion with the MT and a bandpass response from the DI-RR. **f** The measured RF gain and NF of the RF bandpass filter. Maximum RF gain of 1.2 dB and minimum noise figure of 21.8 dB are achieved using the low-biasing MZM. **g** The measured SFDR of the RF bandpass filter at 7 GHz, reaching a high value of 112 dB Hz$^{2/3}$. EDFA: erbium-doped fiber amplifier, MZM: Mach–Zehnder modulator, PC: polarization controller, RF: radio frequency, PD: photodetector, IMD3: third-order intermodulation distortion.

by using the combination of MT and DI-RR circuits in the same photonic chip.

We implement third-order intermodulation distortion (IMD3) cancellation technique using complex multi-order sidebands spectral shaping as we previously reported[31,41,42] (see Supplementary Information E and F). In this experiment, light is sent to a phase modulator (PM) (Fig. 4a) that generates the optical carrier and first and higher order sidebands. The MT is used to convert the PM to aDSB modulation format, and to shape the amplitudes and phases of the optical carrier and multi-order sidebands yielding to IMD3 suppression. The DI-RR is then used to filter the first order sideband to create a phase cancellation RF notch filter. In this way, simultaneous RF notch filter and linearization can be achieved, leading to a significant SFDR enhancement (see Supplementary Information E and F).

A standard SSB RF notch filter without any linearization was used as a benchmark for the proposed linearized RF notch filter. Figure 4b shows the proposed linearized RF notch filter with >45 dB rejection improvement compared to the standard SSB RF notch filter. With the linearization, we achieved 29 dB of IMD3 suppression (Fig. 4c) and 20 dB SFDR improvement (Fig. 4d). With a noise PSD of −164.5 dBm/Hz

and estimated IIP3 of +15 dBm, the standard SSB RF notch filter shows an SFDR of 103 dB Hz$^{2/3}$, while the linearized RF notch filter exhibits a record-high SFDR of 123 dB Hz$^{4/5}$. It is clear from the slope of the measured IMD3 powers that the fifth-order distortion becomes the dominant factor due to cancellation of the third-order distortion. Similar record-high SFDR also can be observed in different frequencies (see Supplementary Information G). These results marked the highest SFDR simultaneously achieved with on-chip signal functionality.

## DISCUSSION

Table 1 summarizes the performance comparison of existing reconfigurable filter circuits. The route to largest number of functions is still provided through MZI-mesh circuits[22]. Virtually all demonstrations, except the application specific RF notch filter reported in ref. [25], show relatively low RF performance, notably characterized by very high NF (>25 dB) and low SFDR (<100 dB Hz$^{2/3}$). These results highlight the challenge in creating multifunctional and high-performance MWP circuits. Our results overcome this challenge by delivering high level of reconfigurability with a variety of functions in combination with high gain, low NF, and ultrahigh SFDR. Importantly, in the context of

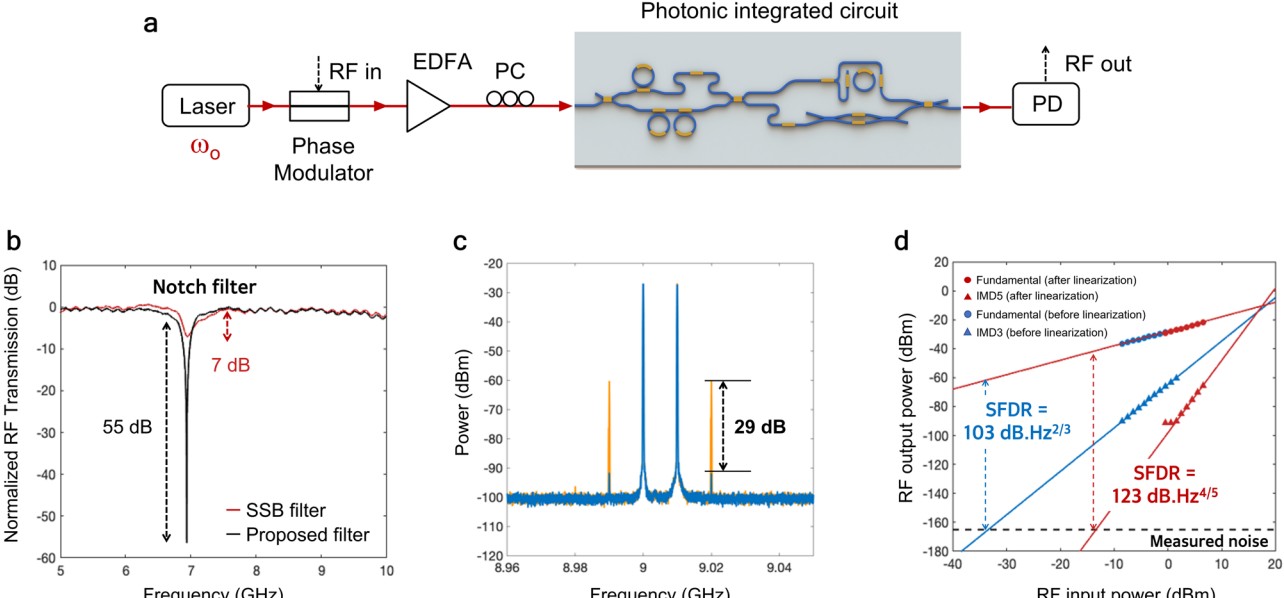

**Fig. 4 | Linearization of RF notch filter. a** Experiment setup of simultaneous RF notch filter with linearization in a phase modulator (PM)-based system. **b** Measured RF notch filter with 55 dB rejection. **c** The measured two-tone RF spectra at the output of the photodetector for the linearized RF notch filter (blue) and single-sideband (SSB) RF notch filter without linearization (yellow) with 29 dB reduction of IMD3 power was achieved. **d** The measured spurious-free dynamic range (SFDR) of the proposed linearized RF notch filter and SSB RF notch filter without linearization at RF frequency of 9 GHz. The proposed linearized RF notch filter has a record-high SFDR of 123 dB Hz$^{4/5}$. EDFA: erbium-doped fiber amplifier, PM: Phase modulator, PC: polarization controller, RF: radio frequency, PD: photodetector, IMD3: third-order intermodulation distortion, IMD5: fifth-order intermodulation distortion.

functional circuits, our demonstrations represent record-low NF and record-high SFDR to date.

At present, our approach enables NF reduction through low biasing of MZM intensity modulator. This translates into moderate SFDR enhancement as seen in Table 1. Our linearization method, on the other hand, is working for PM-based MWP filter where maximum IMD3 suppression is achieved when the optical carrier is partially suppressed (See Supplementary Information D). Due to the limitation in maximum optical amplification in our experimental setup, the measured ultra-high SFDR is still accompanied by relatively low link gain and high NF (Table 1). Achieving the gain, NF, and SFDR advantages simultaneously is feasible in the PM link when higher optical amplification and higher power handling components are used. Because the current linearization method only works in PM-based MWP notch filter, a similar strategy development is desirable for low-biased IM-based MWP filters.

At the current state, the positive RF gain in both IM-based MWP filters is achieved using two external erbium-doped fiber amplifiers (EDFAs). The recent advances in low-noise amplification using erbium-doped waveguide amplifiers in silicon nitride[43] can offer a high-performance integrated MWP system with low weight and reduced footprint.

In summary, we have designed, fabricated, and experimentally demonstrated programmable integrated MWP circuit based on a unique combination of versatile MT and equally versatile DI-RR. We reconfigure the circuit to synthesize an array of RF filtering functions with high RF gain, low NF, and ultrahigh SFDR concurrently. This work opens a new paradigm and an important step to realize programmable integrated MWP circuits with versatile functions, low NF, and ultra-high dynamic range suited for real-life applications.

## Methods
### Silicon nitride circuit fabrication
The waveguides in our circuit are fabricated using LioniX standard TriPleX Asymmetric Double-Stripe (ADS) geometry[44,45]. First, a SiO$_2$ layer is grown from wet thermal oxidation of single-crystal silicon substrate with temperatures equal to or above 1000 °C. Then, low-pressure chemical vapor deposition (LPCVD) is used for the Si$_3$N$_4$ layers and together with the gas tetraethylorthosilicate (TEOS) for the intermediate SiO$_2$ layer. Next, the waveguides are patterned using contact lithography, and processed with dry etching. Last, the waveguides are covered with an additional SiO$_2$ layer through LPCVD TEOS. Because typical top cladding thickness cannot be achieved only by LPCVD TEOS, plasma-enhanced chemical vapor deposition (PECVD) is used to increase the SiO$_2$ top cladding thickness to a total of 8 $\mu$m. Thicker layers can be achieved because of the stress in PECVD SiO$_2$ layers is much less than LPCVD layers.

### Details of programmable microwave photonics experiments
A laser (Pure Photonics PPCL550) with low relative-intensity noise (RIN) of -155 dB/Hz is amplified with a low-noise erbium-doped fiber amplifier (EDFA, Amonics). Then, the output of amplified optical carrier is modulated using a MZM (Thorlabs, LNLVL-IM-Z 40 GHz, $V_{\pi,\mathrm{RF}}$ of 2.2 V at 1 GHz and 3.5 V at 20 GHz) with bias point set at quadrature ($\theta_B = \pi/2$). The MZM is driven by an RF signal from a vector network analyzer (VNA, Keysight P5007A). The output of the MZM then sent to another low-noise EDFA (Amonics) before being injected into a programmable silicon nitride chip (LioniX International BV) fabricated using low-loss TriPleX (Si$_3$N$_4$/SiO$_2$) technology[44,45] with propagation loss of the optical waveguide at 0.1 dB/cm. The chip is tuned using thermo-optic tuning and a custom-made heater controller software, while being stabilized by a thermoelectric cooler (TEC) controller. The processed optical signal is sent to a photodetector (PD, APIC 40 GHz) and the converted RF signal is measured with a VNA, while the RF spectrum analyzer (RFSA, Keysight N9000B) is used to measure the RF filter's noise and linearity.

### Details of linearization experiments
A low RIN laser (Pure Photonics PPCL550) is modulated using a PM (EOSpace 20 GHz) using a sweeping RF signal with −3 dBm power from

**Table 1 | Performance comparison of reconfigurable filter circuits**

| Year | Technology platform | Type of devices | Number of functions | Type of functions | Tuning range (GHz) | Performance enhancement | Gain (dB) | Noise figure (dB) | SFDR (dB Hz$^{2/3}$) |
|---|---|---|---|---|---|---|---|---|---|
| 2006[46] | CMOS | Active B-Cell | 1 | BPF | N/A | No | 0 | 15 | N/A |
| 2008[47] | CMOS | Active inductor | 1 | BPF | N/A | No | 6 | 18 | N/A |
| 2017[18] | InP | Las., Mod., RR, PD | 1 | LPF | 0-6 | No | −20 | N/A | 81.4 |
| 2018[22] | SOI | MZI mesh | 20 | BPF, notch | N/A | No | N/A | N/A | N/A |
| 2017[25] | Si₃N₄ | RR | 1 | Notch | 0–12 | NF:LB | 8 | 15.6 | 116 |
| 2018[48] | SOI | Mod., RR, PD | 1 | BPF | 3–10 | No | −39 | N/A | 92.4 |
| 2018[49] | Si | MEMS | 1 | BPF | 20–40 | No | −1.1 | N/A | N/A |
| 2018[50] | Si | MEMS | 1 | BPF | 5.5–15 | No | −3 | N/A | N/A |
| 2019[51] | As₂S₃ | RR, SBS | 1 | Notch | 0–15 | NF:LB | −10 | 27.1 | 96.5 |
| 2019[52] | InP | Las., Mod., MMI | 7 | BPF, notch IFM, RFG | 8–15 | No | N/A | N/A | N/A |
| 2020[53] | SOI | SBS | 1 | BPF | 4–10 | No | −17 | 56.7 | 90.3 |
| 2020[54] | Si₃N₄ | RR | 1 | BPF | 2–7 | NF:CS | −10 | 27 | N/A |
| 2021[28] | Si₃N₄ | RR | 1 | Notch | 3–10 | NF:CS | 3 | 31 | 100 |
| 2021[29] | SOI | Mod., MT, RR, PD | 2 | Notch, BPF | 5–25 | No | N/A | N/A | N/A |
| 2021[20] | InP+SOI | Las., Mod., RR, PD | 2 | Notch, BPF | 3–25 | No | −28 | 51 | 99.7 |
| 2021[55] | Si₃N₄+As₂S₃ | RR, SBS | 1 | Notch | 2–12 | No | N/A | N/A | 92.2 |
| 2021[56] | Si₃N₄+LiNbO₃ | Mod., RR | 1 | Downconversion | 4–20 | No | −10 | 45 | 105 |
| This work | Si₃N₄ | MT, DI-RR | 6 | Notch | 5–20 | NF:LB | **10** | **15** | 116 |
|  |  |  |  | BPF | 5–20 | NF:LB | 1.2 | **21.8** | 112 |
| This work | Si₃N₄ | MT, DI-RR | 1 | Notch | 6–18 | SFDR:Lin. | −26 | 35 | **123** |

*RR* ring resonator, *PD* photodetector, *Mod* modulator, *MMI* multi-mode interference, *SBS* stimulated Brillouin scattering, *MT* modulation transformer, *DI-RR* double-injection ring resonator, *LPF* low pass filter, *BPF* bandpass filter, *IFM* instantaneous frequency measurement, *PS* phase shifter, *NF* noise figure, *LB* low biasing, *CS* carrier suppression, *SFDR* spurious-free dynamic range, *Lin* linearization, *CMOS* complementary metal-oxide-semiconductor, *RC* resistor–capacitor, *MEMS* microelectromechanical systems.

VNA (Keysight P5007A) for RF notch filter experiment. For the IMD3 suppression experiment, the two-tone RF signal with a power of −0.5 dBm, centered at 9 GHz with a space of 10 MHz from signal generators (Wiltron 69147A and Rohde-Schwarz SMP02) is used to drive the PM. The modulated signal is then amplified by a low-noise EDFA (Amonics), before coupling to a programmable silicon nitride chip (LioniX International BV). The chip is tuned using thermo-optic elements and stabilized by a TEC controller. The processed optical spectrum is sent to a PD (APIC 40 GHz) to retrieve the RF signal and measured using the VNA for the filter's response, and the RFSA (Keysight N9000B) for linearity.

## Data availability
The data that support the plots within this paper and the supplementary materials are available at https://doi.org/10.4121/21207662.

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

## Acknowledgements
This work was supported by Netherlands Organisation for Scientific Research NWO Vidi (15702) and NWO Start Up (740.018.021).

## Author contributions
O.D. and G.L. contributed equally to this work. O.D., G.L., and D.M. developed the concept and proposed the physical system. O.D. designed the photonic circuits, O.D. and G.L. developed and performed numerical simulations. O.D. and G.L. performed the experiments with input from K.Y., R.B., and Y.K. E.K., M.H., and C.R. layout, developed, and fabricated the silicon nitride circuits. D.M., O.D., and G.L. wrote the manuscript with input from Q.T., H.Y. and Y.L. D.M. led and supervised the entire project.

## Competing interests
The authors declare no competing interests.
