## [Peer Review File · Nature Communications]

Ultrahigh Dynamic Range and Low Noise Figure Programmable Integrated Microwave Photonic FilterReviewers' comments:

Reviewer #1 (Remarks to the Author):

This is a good work covering the essences of working principles of integrated microwave photonics. The general readers would find useful information on the concept, implementation, and performance optimization of such approaches for RF functionalities. The authors are well known experts in the field. Their previous work has built up the basis and led to a new contribution to the field reported in this work. The work reported here is sound and complete including both theory and experimental verification and results discussion. It is also clear to see the amount of effort in different forms put in this work. I recommend acceptance for publication. On the other hand, above the reported work, I would like to ask authors to add comments on their options what is the future outlook of microwave photonics? What key technology improvement must be reached for a true advancing? What would be the must-be-microwave photonics applications?

Reviewer #2 (Remarks to the Author):

A programmable integrated microwave photonic (MWP) filter based on integrated SiN waveguide technology is presented. The circuit comprises a modulation transformer (MT) and a double-injected ring resonator (RR). A high rejection notch filter with high RF gain of 10 dB, low noise figure (NF) of 15 dB, and large spurious free dynamic range (SFDR) of 116 dB/Hz is first demonstrated using a low-biasing technique for the off-chip intensity modulator (IM).

Using the same set-up, a passband RF filter transfer function can be obtained by properly reconfiguring the circuit. By replacing the external IM with a PM, a SFDR in excess of 120 dB.Hz is also reported at frequencies outside the RF notch filter, using an on-chip linearization technique previously developed by the authors.

The scheme is conceived for combining the advantages of programmable photonic integration without sacrificing the performance of application-specific circuits. Although, I agree that the work represents a step forward in this sense, I believe that this claim is only partially supported by the results that, in my opinion, seems to provide marginal novelty and/or improvements, as discussed in the following comments.

MAIN COMMENTS

The architecture is not entirely novel, as it follows from previous works from the authors (references [40] [42], [59]), the main novelty seemingly being the inclusion of the DI-MRR (previously discussed for instance in [43]) enabling for switching between the notch and bandpass RF filtering functionality (contrary to [40] where the two functions are obtained using two cascaded ring resonators). Noise figure reduction through low-biased IM is also well-known. A preliminary study of the same circuit discussed in the manuscript has recently appeared in [O. Daulay, "Programmable Integrated Microwave Photonic Filter using a Modulation Transformer and a Double-Injection Ring Resonator." 2021 European Conference on Optical Communication (ECOC). IEEE, 2021] that should be quoted in the manuscript, and in reference [45].

The proposed passband RF filter guarantees a moderate out-of-band rejection in the RF

transmission response of 15 dB in a limited frequency range between 5 and 15 GHz, as reported for instance in Fig. S6(b) of the supplementary material. More importantly, the same figure shows that the filter response degrades when departing from the optimal condition of Fig. 3(e) and gets very noisy when tuned above about 10 GHz. This should be discussed in the manuscript.

Wide tuning range is often regarded as a key feature of MWP filters with respect to electronics-based solutions. However, given the operation at relatively low frequencies, these results should be compared with electronics circuits performing bandpass operation with better performance in terms of rejection and tunability, and operating in the same or even higher range of frequencies.

A couple of examples are provided below:

Z. Yang, D. Psychogiou and D. Peroulis, "Design and Optimization of Tunable Silicon-Integrated Evanescent-Mode Bandpass Filters," *IEEE Trans. Microwave Theory Tech.*, vol. 66, no. 4, pp. 1790-1803, Apr. 2018.

J. Chang, et al., "High Performance, Continuously Tunable Microwave Filters Using MEMS Devices With Very Large, Controlled, Out-of-Plane Actuation," in *Journal of Microelectromechanical Systems*, vol. 27, no. 6, pp. 1135-1147, Dec. 2018, d

Concerning the performance of the RF notch filter using IM, the authors state that "These results represent the best RF gain, noise figure, and SFDR combination for an MWP notch filter". There seems however to be marginal improvements with respect to the results previously reported from the authors in [36] and [37] (RF gain from 8 to 10, NF from 16 to 15, and the same SFDR of 116 dB.Hz)

Positive RF gain is achieved using two EDFAs. For applications in which compactness/low-weight is a primary requirement, what would be the RF gain without optical amplification in the system?

As shown in Fig. 3 and Fig. 4, the extended linearity operation of the notch filter requires replacing the external IM with a PM, which limits the claimed full-reconfigurability of the device. This should be commented in the text.

MINOR COMMENTS

In Figure 3c, the maximum RF gain and minimum NF is achieved for the minimum considered modulator bias voltage of roughly 9 V. What happens for bias values below that first point? Do the two figures of merit degrade or are there other system performance metrics that would be affected at lower values of modulator bias? What is the half-wave voltage of the modulator?

It seems that in order to switch between scheme "d" and "e" of Fig. S5, a dual-drive Mach-Zehnder intensity modulator should be used with additional RF 90 degree hybrid, as the MT accepts at its input a dual-sideband modulated spectrum for the case of scheme "d" and a single-sideband modulated spectrum for the case of scheme "e". Alternatively, a dual-parallel MZM modulator should be used. This should be discussed in the text

What are the dimensions of the fabricated photonic chip?

Few typos around the text (examples):

On page 2: "...which leading to sideband cancellation..." should be "...which leads to sideband cancellation..."

In the caption of Fig. S5: "at the otch frequency" should be "at the notch frequency"

On page 16 of the Supplementary material "which spatially isolated one sideband", should be "which spatially isolates one sideband"

Reviewer #3 (Remarks to the Author):

In this manuscript, the authors proposed a programmable integrated microwave photonic filter with ultrahigh dynamic range and low noise figure. The principle of the proposed scheme was introduced in the supplementary information, and a proof-of-concept experiment was carried out to demonstrate the capability of the proposed microwave photonic filter to improve the performance of microwave photonic links. However, there still exist many problems, and the proposed integrated microwave photonic filter lacks innovation. Based on the above consideration, I do not think the paper is suitable for publication in Nature Communication in the present form. My comments are listed below.

1) The modulation transformer (MT) with three ring resonators topology and the double-injection ring resonator (DI-RR) have been proposed and demonstrated in [Ref. 1, Ref. 2]. In my opinion, this paper is only a combination of these two methods, so the innovation of the proposed programmable integrated microwave photonic filter is not enough to support publication in Nature Communication.

2) In this manuscript, the units of dynamic range are not uniform. "120 dB" is used in the abstract, "120 dB·Hz" is used in the introduction, "116 dB·Hz^{2/3}" and "116 dB·Hz^{2/3}" are used in Fig. 3.

3) Only the notch filter achieves the high linearity, the SFDRs of other filters are still lower than 120 dB·Hz^{2/3}. Therefore, it is inaccurate to claim that "ultra-high dynamic range of > 120 dB·Hz" is achieved.

4) Since the linearization is only for notch filters, not for all filters, it is misleading to directly use "ultrahigh dynamic range" in the title. Meanwhile, this paper only achieves the integration of several passive devices, so "integrated microwave photonic filter" is also inappropriate.

5) From the supplementary information, the tuning range of the bandpass filter is obviously not as wide as the 4-20GHz stated in TABLE I.

6) Only the normalized response is given in the manuscript, and the non-normalized response also should be given, so that the reader can understand the actual insertion loss of the microwave photonic filter.

7) The linearization is realized by the widely-used optical sideband phase and amplitude manipulation, which is usually regarded as a narrowband approach. So, within which frequency range the SFDR can reach 120 dB·Hz^{2/3}? The authors should provide the SDFR results at other frequencies.

8) From the description of the experimental details, the power of the input two-tone signal is 10 dBm, but the power of fundamental and spurious at this input power is not reflected in Fig. 4(d). [Ref. 1] L.-W. Luo, S. Ibrahim, A. Nitkowski et al., "High bandwidth on-chip silicon photonic interleaver," Optics express, vol. 18, no. 22, pp. 23 079–23 087, 2010.

[Ref. 2] R. Cohen, O. Amrani, and S. Ruschin, "Response shaping with a silicon ring resonator via double injection," Nature Photonics, vol. 12, no. 11, pp. 706–712, 2018.

We also thank the reviewers for their valuable comments and their time spent in evaluating this work. In this response, we addressed each comment/question from the reviewers point by point. The reviewer's comments are in **Black**, our responses are in **Blue**, and revisions in the manuscript are in **Red**.

Point-to-point response:

Reviewer: 1

This is a good work covering the essences of working principles of integrated microwave photonics. The general readers would find useful information on the concept, implementation, and performance optimization of such approaches for RF functionalities. The authors are well known experts in the field. Their previous work has built up the basis and led to a new contribution to the field reported in this work. The work reported here is sound and complete including both theory and experimental verification and results discussion. It is also clear to see the amount of effort in different forms put in this work. I recommend acceptance for publication. On the other hand, above the reported work, I would like to ask authors to add comments on their options what is the future outlook of microwave photonics? What key technology improvement must be reached for a true advancing? What would be the must-be-microwave photonics applications?

Reply:

We are grateful for the reviewer's recommendation of publication. In general, the outlook of microwave photonics would be having photonics – electronics co-integration in a single system-on-chip (SoC) or system-in-package (SiP). In particular, the outlook of integrated microwave photonics is having a hybrid-integrated microwave photonic systems containing of a low RIN laser with high output optical power, an ultra-linear modulator with high optical power handling and broad bandwidth, a low loss optical signal processor with mixed architecture (combination of application specific and programmable photonics), and a fast photodetector with high optical power handling and broad bandwidth.

For the proposed architecture, an improved version of modulation transformer (MT) is desirable. The improved version needs to be able to independently tailor the amplitude and the phase of each element in modulation spectrum (optical carrier and sidebands) simultaneously. Then, a new optical circuit design with improved versatility compared to DI-

RR is also desirable with addition of nonlinear optical element, such as stimulated Brillouin scattering (SBS) for improved signal processing resolution.

Reviewer: 2

A programmable integrated microwave photonic (MWP) filter based on integrated SiN waveguide technology is presented. The circuit comprises a modulation transformer (MT) and a double-injected ring resonator (RR). A high rejection notch filter with high RF gain of 10 dB, low noise figure (NF) of 15 dB, and large spurious free dynamic range (SFDR) of 116 dB/Hz is first demonstrated using a low-biasing technique for the off-chip intensity modulator (IM).

Using the same set-up, a passband RF filter transfer function can be obtained by properly reconfiguring the circuit. By replacing the external IM with a PM, a SFDR in excess of 120 dB.Hz is also reported at frequencies outside the RF notch filter, using an on-chip linearization technique previously developed by the authors.

The scheme is conceived for combining the advantages of programmable photonic integration without sacrificing the performance of application-specific circuits. Although, I agree that the work represents a step forward in this sense, I believe that this claim is only partially supported by the results that, in my opinion, seems to provide marginal novelty and/or improvements, as discussed in the following comments.

MAIN COMMENTS

1) The architecture is not entirely novel, as it follows from previous works from the authors (references [40] [42], [59]), the main novelty seemingly being the inclusion of the DI-MRR (previously discussed for instance in [43]) enabling for switching between the notch and bandpass RF filtering functionality (contrary to [40] where the two functions are obtained using two cascaded ring resonators). Noise figure reduction through low-biased IM is also well-known. A preliminary study of the same circuit discussed in the manuscript has recently appeared in [O. Dauly, "Programmable Integrated Microwave Photonic Filter using a Modulation Transformer and a Double-Injection Ring Resonator." 2021 European Conference on Optical Communication (ECOC). IEEE, 2021] that should be quoted in the manuscript, and in reference [45].

Reply:

We disagree with the assessment of this reviewer. Integrated microwave photonics, much like RF electronics, relies on judicious interconnection of known, but reliable components which in turn enables prime performance or functionality that is previously unachievable. This is illustrated by the following examples, recently published in Nature Communications:

- Silicon Brillouin waveguide: S. Gertler et al., *Narrowband microwave-photonic notch filters using Brillouin-based signal transduction in silicon*, **Nat. Commun.** 13(1), 2022.

- Dual integrated lasers: S. Jia et al., Integrated dual-laser photonic chip for high-purity carrier generation enabling ultrafast terahertz wireless communications, **Nat. Commun.** 13, 1388, 2022.
- Mach-Zehnder interferometer delay lines: V. Duarte et al., Modular coherent photonic-aided payload receiver for communications satellites, **Nat. Commun.** 10, 1984, 2019.

Indeed, the inclusion of the DI-RR in our work is very novel in the context of microwave photonics. The interconnection of DI-RR and the versatile modulation transformer (MT) in a low-loss integration platform such as silicon nitride (as opposed to Silicon-on-insulator in Ref [40] in the revised manuscript) can mutually unlock the performance potential and application scope, which has never been attempted before. The combination of all aspects, in this case is a must to achieve the one-of-a-kind combination of performance metrics reported in this work (i.e., programmability, tight integration, and high RF performance).

Moreover, when compared to the previous work mentioned by this reviewer (Ref [40] in the revised manuscript), our work significantly overcomes the limitation in previous MT design, simultaneously achieving high bandwidth (>40 GHz over merely 10 GHz) and ultra-sharp filtering response (3.5 GHz transition band over 7 GHz in previous work) that allows for the demanded access to low RF frequencies. Compared to previously used two cascaded ring resonators, the inclusion of DIRR lead to previously unattainable functionalities in reduced chip area. We also show the first ever complex (phase and amplitude) characterization of DI-RR (Supplementary Information B. Importantly, there was no investigation nor optimization of RF performance in previous works (Ref [40] in the revised manuscript). In fact, our work reports the best RF performance ever reported for a reconfigurable integrated microwave photonic system.

We agree to add the recent ECOC 2021 manuscript in the list of references (Ref [43] in the revised manuscript).

2) The proposed passband RF filter guarantees a moderate out-of-band rejection in the RF transmission response of 15 dB in a limited frequency range between 5 and 15 GHz, as reported for instance in Fig. S6(b) of the supplementary material. More importantly, the same figure shows that the filter response degrades when departing from the optimal condition of Fig. 3(e) and gets very noisy when tuned above about 10 GHz. This should be discussed in the manuscript.

Reply:

We thank the reviewer for the suggestion. We have included new measurement results of the RF bandpass filter with improved filtering extinction ratio and wideband tunability in the main text. The bandpass filter has >20 dB extinction over 5-20 GHz frequency tuning. Through improved control of the measurement parameters, we have maintained the integrity of the passband response over the entire tuning range. Moreover, we have demonstrated that it is possible to have both high link gain and low noise figure in the filter passband. The measured metrics of 1.2 dB (positive/amplified) link gain and 21.8 dB noise figure constitute 11.2 dB gain enhancement and 5.2 dB reduction of noise figure when compared to any previously demonstrated work with next best performance metrics (Ref [54] in the revised manuscript), with nearly 3 times wider tuning range and nearly 13 dB improves SFDR when compared to any reported integrated MWP BPF. Therefore, our results represent the best performance metrics ever for integrated MWP BPF and represent a significant step forward in the field.

These results have now been included in the main text of the revised manuscript, and an excerpt of the figure has been included below.

Revision:

We have updated Fig. 3(e) – 3(g) of the main text with new and significantly improved experimental data.

3) Wide tuning range is often regarded as a key feature of MWP filters with respect to electronics-based solutions. However, given the operation at relatively low frequencies, these results should be compared with electronics circuits performing bandpass operation with better performance in terms of rejection and tunability, and operating in the same or even higher range of frequencies.

A couple of examples are provided below:

Z. Yang, D. Psychogiou and D. Peroulis, "Design and Optimization of Tunable Silicon-Integrated Evanescent-Mode Bandpass Filters," *IEEE Trans. Microwave Theory Tech.*, vol. 66, no. 4, pp. 1790-1803, Apr. 2018.

J. Chang, et al., "High Performance, Continuously Tunable Microwave Filters Using MEMS Devices With Very Large, Controlled, Out-of-Plane Actuation," in *Journal of Microelectromechanical Systems*, vol. 27, no. 6, pp. 1135-1147, Dec. 2018, d

Reply:

We thank the reviewer for the suggestion. We have included the examples cited by the reviewer for comparison in Table I. Indeed, these MEMS-based RF filters can exhibit operation at very high frequencies. The one from Chang *et al.* exhibits similar tuning range as our filter but with higher insertion loss (-3 dB) and does not report the noise figure. The work of Yang *et al.* has larger tuning range 20-40 GHz, but also higher insertion loss (-1.1 dB). But of course, the glaring difference with our results is that these RF filters are limited in the synthesized response, i.e., only bandpass filter. Our results embodied the real strengths of microwave photonics, where a single circuit can be programmed to exhibit various functions and responses, leading to ultra-versatile front end. The tuning range reported here is not by any means a fundamental limitation and is currently limited only by our experimental apparatus. Previously, 1-30 GHz tunable MWP filter with similar principle of operation has been demonstrated (Ref [47] in the revised manuscript).

Revision:

We have updated the Table I in the revised main text with additional comparison data.

TABLE I. Performance Comparison of Programmable Microwave Photonic Circuits

Year	Technology platform	Type of devices	Number of functions	Type of functions	Tuning range (GHz)	Performance enhancement	Gain (dB)	Noise figure (dB)	SFDR (dB · Hz ^{2/3})
2017 [10]	InP	Las., Mod., RR, PD	1	LPF	0-6	No	-20	N/A	81.4
2018 [33]	SOI	MZI mesh	20	BPF, notch	N/A	No	N/A	N/A	N/A
2017 [36]	Si ₃ N ₄	RR	1	Notch	0-12	NF:LB	8	15.6	116
2018 [48]	SOI	Mod., RR, PD	1	BPF	3-10	No	-39	N/A	92.4
2018 [49]	Si	MEMS	1	BPF	20-40	No	-1.1	N/A	N/A
2018 [50]	Si	MEMS	1	BPF	5.5-15	No	-3	N/A	N/A
2019 [51]	As ₂ S ₃	RR, SBS	1	Notch	0-15	NF:LB	-10	27.1	96.5
2019 [52]	InP	Las., Mod., MMI	7	BPF, notch IFM, RFG	8-15	No	N/A	N/A	N/A
2020 [53]	SOI	SBS	1	BPF	4-10	No	-17	56.7	90.3
2020 [54]	Si ₃ N ₄	RR	1	BPF	2-7	NF:CS	-10	27	N/A
2021 [39]	Si ₃ N ₄	RR	1	Notch	3-10	NF:CS	3	31	100
2021 [40]	SOI	Mod., MT, RR, PD	2	Notch, BPF	5-25	No	N/A	N/A	N/A
2021 [12]	InP+SOI	Las., Mod., RR, PD	2	Notch, BPF	3-25	No	-28	51	99.7
2021 [55]	Si ₃ N ₄ +As ₂ S ₃	RR, SBS	1	Notch	2-12	No	N/A	N/A	92.2
2021 [56]	Si ₃ N ₄ +LiNbO ₃	Mod., RR	1	Downconversion	4-20	No	-10	45	105
(this work)	Si ₃ N ₄	MT, DI-RR	6	Notch	5-20	NF:LB	10	15	116
(this work)	Si ₃ N ₄	MT, DI-RR	1	BPF	5-20	NF:LB	1.2	21.8	112
(this work)	Si ₃ N ₄	MT, DI-RR	1	Notch	6-18	SFDR:Lin.	-26	35	123

RR: ring resonator, PD: photodetector, Mod: modulator, MMI: multi-mode interference, SBS: stimulated Brillouin scattering, MT: modulation transformer, DI-RR: double-injection ring resonator, LPF: low pass filter, BPF: bandpass filter, IFM: instantaneous frequency measurement, PS: phase shifter, NF: noise figure, LB: low biasing, CS: carrier suppression, SFDR: spurious-free dynamic range, Lin: linearization, MEMS: microelectromechanical systems.

4) Concerning the performance of the RF notch filter using IM, the authors state that “These results represent the best RF gain, noise figure, and SFDR combination for an MWP notch filter”. There seems however to be marginal improvements with respect to the results previously reported from the authors in [36] and [37] (RF gain from 8 to 10, NF from 16 to 15, and the same SFDR of 116 dB.Hz)

Reply:

We stand by our claim that the reported values are the best performance combination for any MWP notch filter. The fact that we achieved these improved values with wider tuning range, and much improved programmability should be acknowledged. We should note that the reported link performance in previous work is achieved using very simple integrated photonic devices with low interconnection loss, i.e., only a ring resonator; in contrast, here we use much more complex integrated photonic circuits, but achieved comparable or even better performance. In addition, these record values for the notch filter are obtained using the same system as the one demonstrating a record performance for bandpass filter. In our experiments we used a low

half-wave voltage modulator, with V_{π} of 2.2 V. This leads to the improvements in link gain and noise figure of the filter.

5) *Positive RF gain is achieved using two EDFAs. For applications in which compactness/low-weight is a primary requirement, what would be the RF gain without optical amplification in the system?*

Reply:

We thank the reviewer for the question. At present, RF performance without amplification is way below our demonstrated performance. For example, non-amplified filter with gain = -28 dB, NF = 51 dB and SFDR of 99.7 dB.Hz^{2/3} has been reported (Ref [12] in the revised manuscript) has been reported. In our case, the unamplified and unoptimized RF gain was -56.5 dB. However, adding amplifiers to the system need not to add significant weight and footprint. Mini and micro-EDFAs technology has been used in microwave photonics systems. Moreover, recent advances in low-noise amplification using erbium-doped waveguide amplifiers in silicon nitride [1] can offer high performance integrated MWP system with low weight and footprint.

Reference:

[1] Y. Liu *et al.*, A photonic integrated circuit-based erbium-doped amplifier, **Science**, 376, 1309-1313, 2022.

Revision:

We have added the statement about amplification in discussion part of main text.

At current state, the positive RF gain in both IM-based MWP filters is achieved using two external erbium-doped fiber amplifiers (EDFAs). The recent advances in low-noise amplification using erbium-doped waveguide amplifiers in silicon nitride [63] can offer high performance integrated MWP system with low weight and reduced footprint.

As shown in Fig. 3 and Fig. 4, the extended linearity operation of the notch filter requires replacing the external IM with a PM, which limits the claimed full-reconfigurability of the device. This should be commented in the text.

Reply:

We understand the point of view of the reviewer. Indeed, that the linearization technique we present here thus far works only for PM links. This is why we didn't claim the linearized performance together with the IM-based system (see Table 1, where we separate the results in two lines). What we claim is that the same circuit can carry out both the IM-based and PM-based performance enhancements, therefore highlighting the reconfigurability of the circuit. We have explained this in the revised manuscript as the following.

Revision:

We have added this limitation in discussion part of main text.

Our linearization method, on the other hand, is working for PM-based MWP filter where maximum IMD3 suppression is achieved when the optical carrier is partially suppressed (See Supplementary Information D). Due to the limitation in maximum optical amplification in our experimental setup, the measured ultra-high SFDR is still accompanied by relatively low link gain and high NF (Table 1). Achieving the gain, NF, and SFDR advantages simultaneously is feasible in the PM link when higher optical amplification and higher power handling components are used. Because current linearization method only works in PM-based MWP notch filter, a similar strategy development is desirable for low-biased IM-based MWP filters.

MINOR COMMENTS

In Figure 3c, the maximum RF gain and minimum NF is achieved for the minimum considered modulator bias voltage of roughly 9 V. What happen for bias values below that first point? Do the two figures of merit degrade or are there other system performance metrics that would be affected at lower values of modulator bias? What is the half-wave voltage of the modulator?

Reply:

We thank the reviewer for the question. When the bias voltage is tuned to lower voltage (< 9.3 V), the RF gain degrades and the NF increases. This transition can be seen in the updated Fig. 3(c) and (f) in the main text. The RF half-wave voltage of the IM according to the datasheet is 2.2 V at 1 GHz and 3.5 V at 20 GHz.

Revision:

We have updated the measured RF performance of the notch and bandpass filter in Fig. 3(c) and 3(f) of the revised main text.

It seems that in order to switch between scheme “d” and “e” of Fig. S5, a dual-drive Mach-Zehnder intensity modulator should be used with additional RF 90-degree hybrid, as the MT accepts at its input a dual-sideband modulated spectrum for the case of scheme “d” and a single-sideband modulated spectrum for the case of scheme “e”. Alternatively, a dual-parallel MZM modulator should be used. This should be discussed in the text

Reply:

We thank the reviewer for the positive assessment of our work. The switch between scheme “d” and “e” of Fig. S5 can be happened using an external bandpass filter to completely cancel one sideband in an IM spectrum. The authors added new setup used to switch between scheme “d” and “e” of Fig. S5.

Revision:

We updated with additional setup used to switch between scheme “d” and “e” of Fig. S5 and the description of extended experiment in Supplementary Information C.

What are the dimensions of the fabricated photonic chip?

Reply:

We thank the reviewer for the question. The fabricated photonic chip containing of multiple circuit with dimension of 8 x 16 mm with the proposed circuit occupy area with dimension of 3.2 x 6.8 mm.

Few typos around the text (examples):

On page 2: "...which leading to sideband cancellation..." should be "...which leads to sideband cancellation..."

In the caption of Fig. S5: "at the otch frequency" should be "at the notch frequency" On page 16 of the Supplementary material "which spatially isolated one sideband", should be "which spatially isolates one sideband"

Reply:

We thank the reviewer for the input of our work. We have corrected these errors in our revised manuscript.

Revision:

We exemplify the intensity modulation-to-phase modulation (IM-PM) conversion by inverting the phase of one sideband by π , which leads to sideband cancellation with 62 dB extinction in direct photodetection (Fig. 2b).

Cancellation notch filter. MT is used to create asymmetric dual-sideband modulation conversion while DI-RR shows 5 dB-deep notch response. Destructive interference at the notch frequency amplifies the RF notch filter response to 58 dB.

An IM signal is sent to pass through a spectral de-interleaver in the MT, which spatially isolates one sideband in the spectrum from another sideband and optical carrier.

Reviewer: 3

In this manuscript, the authors proposed a programmable integrated microwave photonic filter with ultrahigh dynamic range and low noise figure. The principle of the proposed scheme was introduced in the supplementary information, and a proof-of-concept experiment was carried out to demonstrate the capability of the proposed microwave photonic filter to improve the performance of microwave photonic links. However, there still exist many problems, and the proposed integrated microwave photonic filter lacks innovation. Based on the above consideration, I do not think the paper is suitable for publication in Nature Communication in the present form. My comments are listed below.

1) The modulation transformer (MT) with three ring resonators topology and the double-injection ring resonator (DI-RR) have been proposed and demonstrated in [Ref. 1, Ref. 2]. In my opinion, this paper is only a combination of these two methods, so the innovation of the proposed programmable integrated microwave photonic filter is not enough to support publication in Nature Communication.

Reply:

We disagree with the assessment of this reviewer. Integrated microwave photonics, much like RF electronics, relies on judicious interconnection of known, but reliable components which in turn enables prime performance or functionality that is previously unachievable. This is illustrated by the following examples, recently published in Nature Communications:

- Silicon Brillouin waveguide: S. Gertler et al., *Narrowband microwave-photonic notch filters using Brillouin-based signal transduction in silicon*, **Nat. Commun.** 13(1), 2022.
- Dual integrated lasers: S. Jia et al., *Integrated dual-laser photonic chip for high-purity carrier generation enabling ultrafast terahertz wireless communications*, **Nat. Commun.** 13, 1388, 2022.
- Mach-Zehnder interferometer delay lines: V. Duarte et al., *Modular coherent photonic-aided payload receiver for communications satellites*, **Nat. Commun.** 10, 1984, 2019.

The [Ref. 1] only discussed about an optical circuit called spectral de-interleaver. Such circuit is only a partial component in a modulation transformer (MT) with additional two channels containing of a tunable attenuator, a phase shifter, a delay line, and a combiner at the output of these two channels. The MT is properly explained in supplementary material. The combination of a spectral de-interleaver and a DIRR is **NOT** equal to the combination of a MT and a DI-RR proposed in this article.

The inclusion of the DI-RR in our work is very novel in the context of microwave photonics. The interconnection of DI-RR and the versatile modulation transformer (MT) in a low-loss integration platform such as silicon nitride (as opposed to Silicon-on-insulator in Ref [40] in the revised manuscript) has never been attempted before. The combination of all aspects, in this case is a must to achieve the one-of-a-kind combination of performance metrics reported in this work (i.e., programmability, tight integration, and high RF performance).

2) *In this manuscript, the units of dynamic range are not uniform. “120 dB” is used in the abstract, “120 dB.Hz” is used in the introduction, “116 dB.Hz^{2/3}” and “116 dB.Hz^{2/3}” are used in Fig. 3.*

Reply:

We thank the reviewer for the critical feedback. We have used uniform expression of dB.Hz^{2/3} throughout our manuscript.

3) *Only the notch filter achieves the high linearity, the SFDRs of other filters are still lower than 120 dB.Hz^{2/3}. Therefore, it is inaccurate to claim that “ultra-high dynamic range of > 120 dB.Hz” is achieved.*

Reply:

We aim to clarify that the SFDR of 112 – 116 dB.Hz^{2/3} achieved for RF bandpass and RF notch filters in this work is still very high when compared to previous results. The only RF notch filter with SFDR of > 100 dB.Hz^{2/3} was reported in Ref [36] in the revised manuscript, which is a work led by the senior author of our paper. This comparison rings even more true when

considering RF bandpass filter, with the highest previously reported SFDR of nearly 13 dB lower than our results ($99.7 \text{ dB}\cdot\text{Hz}^{2/3}$) in Ref [12] in the revised manuscript. The ultra-high dynamic range of $> 120 \text{ dB}\cdot\text{Hz}^{2/3}$ shown in this article is a proof concept of on-chip linearization and simultaneous RF notch filter using a single photonic chip.

To avoid confusion, we have revised the abstract and introduction as

Abstract

...

Here, we report, for the first time, a multi-functional photonic integrated circuit that enables programmable filtering functions with record-high performance. We demonstrate a switchable filter function with record-low noise figure and a RF notch filter with ultra-high dynamic range.

...

Introduction

...

In this work, we demonstrate a programmable integrated MWP circuit with a unique combination of a versatile MT device and a DI-RR, realized in a low-loss silicon nitride platform. With this circuit, we show for the first time, an array of RF filters in three different scenarios simultaneously with record-low noise figure for RF notch and RF bandpass filter, achieved using low-biasing technique [36, 37] in an intensity modulator (IM)-based system, and ultra-high dynamic range for RF notch filter, achieved using on-chip linearization in a phase modulator (PM)-based system.

...

4) Since the linearization is only for notch filters, not for all filters, it is misleading to directly use “ultrahigh dynamic range” in the title. Meanwhile, this paper only achieves the integration of several passive devices, so “integrated microwave photonic filter” is also inappropriate.

Reply:

We believe that all the dynamic range values reported throughout this paper, using both the IM and PM-based enhancement fall in the category of high dynamic range. To put it in the context, these values are more than an order of magnitude higher than previously reported (except for Ref [36] in the revised manuscript, which was a work led by the senior author of this paper). The linearized link indeed shows ultra-high dynamic range with SFDR of $123 \text{ dB}\cdot\text{Hz}^{4/5}$.

Regarding the terminology integrated microwave photonic filter, we never claimed that the device is fully integrated, where the lasers, modulators, detectors, and signal processor are either hybrid/heterogeneously integrated. Nevertheless, we believe, our system is still an integrated microwave photonic system by definition, see for example integrated microwave photonic papers (Ref [6] and Ref [7] in the revised manuscript). Indeed, that it is important to achieve high performance in all integrated system. But in reality, the performance of such systems is much poorer when compared to our results (see Ref [10] and Ref [12] in the revised manuscript as examples). The highest gain, noise figure and SFDR are 39 dB, 36 dB, and 16 dB lower, respectively, than our reported results here. So, our work here serves as evidence that

a partially integrated MWP system can achieve high performance and address a unique area/metric that has never been demonstrated before, and therefore constitute a significant leap in the field.

5) From the supplementary information, the tuning range of the bandpass filter is obviously not as wide as the 4-20GHz, stated in TABLE I.

Reply:

We thank the reviewer for the critical feedbacks to the tuning range of the bandpass filter. We added new experiment results covering the tuning range of the bandpass filter and updated the Table I in main text.

Revision:

We have updated the measured performance of the bandpass filter in Fig. 3(e)-(g) and Table I of the revised main text.

TABLE I. Performance Comparison of Programmable Microwave Photonic Circuits

Year	Technology platform	Type of devices	Number of functions	Type of functions	Tuning range (GHz)	Performance enhancement	Gain (dB)	Noise figure (dB)	SFDR (dB · Hz ^{2/3})
2017 [10]	InP	Las., Mod., RR, PD	1	LPF	0-6	No	-20	N/A	81.4
2018 [33]	SOI	MZI mesh	20	BPF, notch	N/A	No	N/A	N/A	N/A
2017 [36]	Si ₃ N ₄	RR	1	Notch	0-12	NF:LB	8	15.6	116
2018 [48]	SOI	Mod., RR, PD	1	BPF	3-10	No	-39	N/A	92.4
2018 [49]	Si	MEMS	1	BPF	20-40	No	-1.1	N/A	N/A
2018 [50]	Si	MEMS	1	BPF	5.5-15	No	-3	N/A	N/A
2019 [51]	As ₂ S ₃	RR, SBS	1	Notch	0-15	NF:LB	-10	27.1	96.5
2019 [52]	InP	Las., Mod., MMI	7	BPF, notch, IFM, RFG	8-15	No	N/A	N/A	N/A
2020 [53]	SOI	SBS	1	BPF	4-10	No	-17	56.7	90.3
2020 [54]	Si ₃ N ₄	RR	1	BPF	2-7	NF:CS	-10	27	N/A
2021 [39]	Si ₃ N ₄	RR	1	Notch	3-10	NF:CS	3	31	100
2021 [40]	SOI	Mod., MT, RR, PD	2	Notch, BPF	5-25	No	N/A	N/A	N/A
2021 [12]	InP+SOI	Las., Mod., RR, PD	2	Notch, BPF	3-25	No	-28	51	99.7
2021 [55]	Si ₃ N ₄ +As ₂ S ₃	RR, SBS	1	Notch	2-12	No	N/A	N/A	92.2
2021 [56]	Si ₃ N ₄ +LiNbO ₃	Mod., RR	1	Downconversion	4-20	No	-10	45	105
(this work)	Si ₃ N ₄	MT, DI-RR	6	Notch	5-20	NF:LB	10	15	116
				BPF	5-20	NF:LB	1.2	21.8	112
(this work)	Si ₃ N ₄	MT, DI-RR	1	Notch	6-18	SFDR:Lin.	-26	35	123

RR: ring resonator, PD: photodetector, Mod: modulator, MMI: multi-mode interference, SBS: stimulated Brillouin scattering, MT: modulation transformer, DI-RR: double-injection ring resonator, LPF: low pass filter, BPF: bandpass filter, IFM: instantaneous frequency measurement, PS: phase shifter, NF: noise figure, LB: low biasing, CS: carrier suppression, SFDR: spurious-free dynamic range, Lin: linearization, MEMS: microelectromechanical systems.

6) Only the normalized response is given in the manuscript, and the non-normalized response also should be given, so that the reader can understand the actual insertion loss of the microwave photonic filter.

Reply:

We thank the reviewer for the critical feedbacks to the manuscript. We added the non-normalized RF response of both RF notch and RF bandpass filter in Fig.3(b) and 3(e) of the main text.

7) The linearization is realized by the widely used optical sideband phase and amplitude manipulation, which is usually regarded as a narrowband approach. So, within which frequency range the SFDR can reach 120 dB·Hz^{2/3}? The authors should provide the SFDR results at other frequencies.

Reply:

We thank the reviewer for the critical feedbacks to the manuscript. We have added extensive experiment results of SFDR after linearization at other frequencies in Supplementary Information F. The ultra-high SFDR > 120 dB·Hz^{2/3} is maintained at 8, 9, 10, and 16 GHz, showing the broadband nature of our results.

Revision:

We have updated the measured SFDR post linearization at different frequencies in supplementary information F.

To further characterize the performance of our proposed linearized RF notch filter, we extended our measurements of IMD3 suppression and SFDR when the two-tone test frequency and the notch filter frequency was tuned separately. We first fixed the notch frequency at 12 GHz and performed two tone measurements at 8 GHz, 9 GHz, 10 GHz, and 16 GHz. The results of the IMD3 suppression are shown in Fig. S7. It is clear that the IMD3 terms are greatly suppressed for more than 28 dB in all of these two-tone frequencies.

The results of the SFDR at different frequencies are shown in Fig. S8. In all the cases, SFDR of more than $122 \text{ dB} \cdot \text{Hz}^{2/3}$ are observed with improvements around 20 dB compared with nonlinearized states.

Fig. S7. Third-order intermodulation distortion (IMD3) suppression at various two-tone frequencies with notch response at 12 GHz. (a) two-tone signal at 8 GHz (b) two-tone signal at 9 GHz (c) two-tone signal at 10 GHz (d) two-tone signal at 16 GHz.

Fig. S8. Spurious-free dynamic range (SFDR) measurements at various two-tone frequencies with notch response at 12 GHz. (a) SFDR at 8 GHz (b) SFDR at 9 GHz (c) SFDR at 10 GHz (d) SFDR at 16 GHz. IMD3: third-order intermodulation distortion, IMD5: fifth-order intermodulation distortion.

8) From the description of the experimental details, the power of the input two-tone signal is 10 dBm, but the power of fundamental and spurious at this input power is not reflected in Fig. 4(d).

Reply:

We thank the reviewer for the feedback about two-tone signal. The input RF power of two-tone signal to the PM is 8 dBm. We updated the number in main text.

REVIEWER COMMENTS

Reviewer #1 (Remarks to the Author):

My comments have been addressed. I recommend acceptance for publication.

Reviewer #2 (Remarks to the Author):

The authors did good efforts in replying to my comments. As a personal opinion, I believe that the performance of the bandpass functionality for the proposed reconfigurable microwave photonics (MWP) filter is below that of state-of-the-art electronics components. A fair comparison with the MEMS-based devices discussed in refs. [49] and [50] of the revised manuscript should account that the RF loss of 1 and 3 dB, respectively, are obtained in fully passive devices, whereas two power-consuming erbium-doped fiber amplifiers are required for positive RF gain in the proposed MWP approach. At the same time, the MEMS-based filters typically can provide larger rejection, faster roll-off and improved passband flatness (which are parameters that are not considered in TABLE I). The claimed advantage of reconfigurability can be handled in electronics-domain with digitally controlled filter banks and switches, although I agree that this is not an optimal solution. There is also still a concern about the moderate improvement in terms of RF gain and noise figure (NF) (of 2 and 1 dB, respectively) compared with authors' previous works (see refs [36] and [37]) since, as stated by the authors in the answer to comment 4, the improvement seems to be partially ascribed to the use of an external modulator with lower half-wave voltage.

However, in view of the improved quality of the manuscript with new and more accurate measurement results, and in recognizing the advancement in the field brought by the proposed approach in terms of programmable functionality, the paper can be considered for publication in Nature Communications. Below my new comments:

1) In the answer to Comment 3, the authors state that MWP filter tuning in the 1-30 GHz range has been achieved in a previous circuit with similar operation principle. However, the MWP filter discussed in the quoted reference [47] of the revised manuscript is based on stimulated Brillouin scattering (SBS) effect, for which the highest operating frequency is not limited by the free-spectral range (FSR) of microring-resonator (MRRs)-based filtering elements, as in the proposed approach. On the other hand, as discussed in the supplementary material of the original manuscript, the lowest operating frequency of this scheme is 5 GHz, due to the leakage at low modulation frequencies between the two output ports of the spectral de-interleaver in the modulation transformer (MT). This makes the operating frequency range 15 GHz (5-20 GHz, as reported in Table I), and the two statements on page 3 of the revised manuscript claiming that the filter is tunable over 20 GHz should be corrected accordingly.

2) At which frequency is measured the RF gain of the MWP notch filter in Fig. 3c? How does the maximum RF gain scale over the 5-20 GHz filter operating range?

3) In answering my comment about the half-wave voltage of the intensity modulator, the authors indicated the nominal values of 2.2 V and 3.5 V at 1 GHz and 20 GHz, respectively, but missed to include this information in the revised manuscript. Please, add the details of the half-wave voltage for the employed modulator within the main text.

4) Few typos around:

(Abstract): We demonstrate a switchable filter functions...

(Page 5) Similar record-high SFDR also observed in different frequencies (see Supplementary Information F).

(Methods Section) Because typical top cladding thickness cannot achieved only by LPCVD TEOS...

(Methods Section) A thicker layers can be achieved...

Reviewer #3 (Remarks to the Author):

I appreciate the authors taking the effort in answering the questions raised by both the reviewers in previous report. They provided new information to highlight the advantages of their work, which makes it more solid. However, the answers still cannot convince me to recommend this manuscript for publication in Nature Communications:

1) Although the authors give more words to explain their innovation, but it still cannot change the fact that the entire chip is very simple, and the ideas of modulation transformation and noise figure reduction, linearity improvement are reported before. I read the References the authors used to reply the first question from Reviewer 2, from my opinion, the chips reported in these References, e.g., the SBS waveguide, laser chip and the satellite payload chip, are novel in terms of the chip itself or its application systems, which clearly shows the levels of the chip and the system demonstration. However, I still cannot see such distinguished novelty from this paper.

2) The comparison with the electrical filters is not reasonable. Although NF and SFDR are not measured in Ref. [49] and Ref. [50], I believe they have better performances than the photonic filter, since they do not involve EO & OE conversions and active amplification. The advantages of the proposed filter are the relatively higher gain and reconfigurability. However, the gain is realized by two EDFAs which will sacrifice the analog performance. For the reconfigurability, I cannot clearly see (or find from the manuscript) the necessity of such a filter. Therefore, the significance of the work and the impact on the researchers in both MWP and microwave communities are not clear.

3) From the Method, the power of the input two-tone RF signal is 8 dBm. According to Fig. 4(c), the power of the fundamental components are nearly -30 dBm, which cannot agree with the -26 dB claimed in Table I and the results shown in Fig. 4d. Please check it.

4) Although SFDRs at different frequencies are measured, it would be more convincing to directly input a wideband RF signal (for example, a QAM signal) to demonstrate its high dynamic range after filtering.

Ultrahigh Dynamic Range and Low Noise Figure Programmable Integrated Microwave Photonic Filter: 2nd Response to Reviewers' comments

NCOMMS-22-10585

We thank the reviewers for their valuable comments and their time spent in evaluating this work. In this response, we addressed each comment/question from the reviewers point by point. The reviewer's comments are in Black, our responses are in Blue, and revisions in the manuscript are in Red

Point-to-point response:

Reviewer: 1

My comments have been addressed. I recommend acceptance for publication.

Reply:

We thank the reviewer for the positive assessment of our paper.

Reviewer: 2

The authors did good efforts in replying to my comments. As a personal opinion, I believe that the performance of the bandpass functionality for the proposed reconfigurable microwave photonics (MWP) filter is below that of state-of-the-art electronics components. A fair comparison with the MEMS-based devices discussed in refs. [49] and [50] of the revised manuscript should account that the RF loss of 1 and 3 dB, respectively, are obtained in fully passive devices, whereas two power-consuming erbium-doped fiber amplifiers are required for positive RF gain in the proposed MWP approach. At the same time, the MEMS-based filters typically can provide larger rejection, faster roll-off and improved passband flatness (which are parameters that are not considered in TABLE I). The claimed advantage of reconfigurability can be handled in electronics-domain with digitally controlled filter banks and switches, although I agree that this is not an optimal solution. There is also still a concern about the moderate improvement in terms of RF gain and noise figure (NF) (of 2 and 1 dB, respectively) compared with authors' previous works (see refs [36] and [37]) since, as stated by the authors in the answer to comment 4, the improvement seems to be partially ascribed to the use of an external modulator with lower half-wave voltage.

However, in view of the improved quality of the manuscript with new and more accurate measurement results, and in recognizing the advancement in the field brought by the proposed approach in terms of programmable functionality, the paper can be considered for publication in Nature Communications.

Reply:

We thank the reviewer for **the positive recommendation** and the highly valuable perspectives given for our paper.

We agree with the reviewer on the fact that the MEMS-based filter can exhibit a lower noise figure (determined by the passive circuit insertion loss) while the MWP filters can be considered 'active', because the system requires lasers, electro-optic modulators, optical amplifiers, and photodetectors that actively consumes power. These optical components usually contribute to more noise, higher losses, and thus higher noise figure to the system. However, there are number of reports about active bandpass filter that show **comparable** performance with our work because of the usage of active-RC [1], active capacitance [2] or active inductor [3-5] in the circuit.

Despite the > 10 dB noise figures of reported MWP filters, it is fundamentally feasible to achieve single-digit dB level with lower-RIN lasers, high-efficiency modulators and photodetectors, and low-loss photonic interconnects. Thus, reducing the noise figure of MWP filters will be realized based on the advances and innovation of these optical components, which has been continuously driven by the need for high-performance devices not only for optical communications but also for optical signal processing and computing.

References:

[1] B. Wu *et al.*, A 40 nm CMOS Derivative-Free IF Active-RC BPF With Programmable Bandwidth and Center Frequency Achieving Over 30 dBm IIP3, **IEEE Journal of Solid-State Circuits**, 50(8), 1772-1784, 2015.

[2] D. Colaiuda *et al.*, A Second Order 1.8-1.9 GHz Tunable Active Band-Pass Filter with Improved Noise Performance, **Electronics**, 11(2781), 1-10, 2022.

[3] Z. Gao *et al.*, A Fully Integrated CMOS Active Bandpass Filter for Multiband RF Front-Ends, **IEEE Transactions on Circuits and Systems II: Express Briefs**, 55(8), 718-722, 2008.

[4] V. Kumar *et al.*, A 2.5 GHz Low Power, High-Q, Reliable Design of Active Bandpass Filter, **IEEE Transactions on Device and Materials Reliability**, 17(1), 229-244, 2017.

[5] R. Mehra *et al.*, Reliable and Q-Enhanced Floating Active Inductors and Their Application in RF Bandpass Filter, **IEEE Access**, 6, 48181-48194, 2018.

Below my new comments:

1) In the answer to Comment 3, the authors state that MWP filter tuning in the 1-30 GHz range has been achieved in a previous circuit with similar operation principle. However, the MWP filter discussed in the quoted reference [47] of the revised manuscript is based on stimulated Brillouin scattering (SBS) effect, for which the highest operating frequency is not limited by the free-spectral range (FSR) of microring-resonator (MRRs)-based filtering elements, as in the proposed approach. On the other hand, as discussed in the supplementary material of the original manuscript, the lowest operating frequency of this scheme is 5 GHz, due to the leakage at low modulation frequencies between the two output ports of the spectral de-interleaver in

the modulation transformer (MT). This makes the operating frequency range 15 GHz (5-20 GHz, as reported in Table I), and the two statements on page 3 of the revised manuscript claiming that the filter is tunable over 20 GHz should be corrected accordingly.

Reply:

We agree with the comments of the reviewer and have corrected the statement in page 3 to 15 GHz.

Revision:

The RF notch filter central frequency can be tuned from 5 to 20 GHz (Fig. 3b) with maximum RF gain of more than 0 dB over the frequency range (see Supplementary Information D).

RF bandpass filter with 20 dB rejection, with up to 15 GHz tuning range (from 5 GHz to 20 GHz) (Fig. 3e), limited by the roll-off and the dispersion of the spectral de-interleaver, notably at the transition band (see Supplementary Information A). This tuning range can be feasibly increased using improved design of (de)-interleaver with faster roll-off and a larger FSR.

2) *At which frequency is measured the RF gain of the MWP notch filter in Fig. 3c? How does the maximum RF gain scale over the 5-20 GHz filter operating range?*

Reply:

The RF gain of MWP notch filter in Fig. 3c is measured at frequency of 1 GHz. The RF link gain stays above 0 dB over the entire frequency range of 5-20 GHz. Below we list the RF gain at various frequencies:

Freq. (GHz)	RF gain (dB)	Freq. (GHz)	RF gain (dB)	Freq. (GHz)	RF gain (dB)	Freq. (GHz)	RF gain (dB)
5	4.23	9	2.60	13	2.71	17	1.61
6	4.01	10	2.39	14	2.05	18	1.19
7	3.62	11	2.16	15	2.60	19	1.14
8	2.87	12	2.72	16	2.11	20	1.71

Revision:

We have added the list and graph of the maximum RF gain of MWP notch filter over the tuning range (5-20 GHz) in the Supplementary Information D as follows,

Main text:

The RF notch filter central frequency can be tuned from 5 to 20 GHz (Fig. 3b) with maximum RF gain of more than 0 dB over the frequency range (see Supplementary Information D).

Supplementary Information:

The maximum RF gain of the RF notch filter stays above 0 dB over the entire frequency range of 5-20 GHz. Fig. S6 shows the RF gain at various frequencies.

TABLE S1. The maximum RF Gain of the RF notch filter over the entire frequency range

Freq. (GHz)	RF Gain (dB)	Freq. (GHz)	RF Gain (dB)	Freq. (GHz)	RF Gain (dB)	Freq. GHz	RF Gain (dB)
5	4.23	9	2.60	13	2.71	17	1.61
6	4.01	10	2.39	14	2.05	18	1.19
7	3.62	11	2.16	15	2.60	19	1.14
8	2.87	12	2.72	16	2.11	20	1.71

Fig. S6. The RF gain of the RF notch filter. Plot of the maximum RF gain of the RF notch filter over the entire frequency range.

3) In answering my comment about the half-wave voltage of the intensity modulator, the authors indicated the nominal values of 2.2 V and 3.5 V at 1 GHz and 20 GHz, respectively, but missed to include this information in the revised manuscript. Please, add the details of the half-wave voltage for the employed modulator within the main text.

Reply:

We have added the modulator half-wave voltage information in the revised manuscript.

Revision:

(Thorlabs, LNLVL-IM-Z 40 GHz, $V_{\pi,RF}$ of 2.2 V at 1 GHz and 3.5 V at 20 GHz)

4) Few typos around:

(Abstract): *We demonstrate a switchable filter functions ...*

(Page 5) *Similar record-high SFDR also observed in different frequencies (see Supplementary Information F).*

(Methods Section) *Because typical top cladding thickness cannot achieved only by LPCVD TEOS...*

(Methods Section) *A thicker layers can be achieved...*

Reply:

We thank the reviewer for careful reading of our manuscript, we have revised the manuscript accordingly.

Revision:

(Abstract) *We demonstrate reconfigurable filter functions...*

(Page 5) *Similar record-high SFDR also can be observed in different frequencies.*

(Methods Section) *Because typical top cladding thickness cannot be achieved only by LPCVD TEOS...*

(Methods Section) *Thicker layers can be achieved...*

Reviewer: 3

I appreciate the authors taking the effort in answering the questions raised by both the reviewers in previous report. They provided new information to highlight the advantages of their work, which makes it more solid. However, the answers still cannot convince me to recommend this manuscript for publication in Nature Communications:

Reply:

We thank the reviewer for appreciating our work and efforts. In what follows, we will try to address the raised concerns and add suggested clarifications.

1) Although the authors give more words to explain their innovation, but it still cannot change the fact that the entire chip is very simple, and the ideas of modulation transformation and noise figure reduction, linearity improvement are reported before. I read the References the authors used to reply the first question from Reviewer 2, from my opinion, the chips reported in these References, e.g., the SBS waveguide, laser chip and the satellite payload chip, are novel in terms of the chip itself or its application systems, which clearly shows the levels of the chip and the system demonstration. However, I still cannot see such distinguished novelty from this paper.

Reply:

We respectfully disagree that the simplicity of the circuit should be interpreted as lack of novelty in our work. On contrary, we believe that we have demonstrated that such high performance and programmability is indeed accessible through relatively simple chip topology which is previously unachievable.

We agree that individually, modulation transformation, noise figure reduction, and linearization have been achieved. This is precisely the key breakthrough in our work, that we **simultaneously** achieve these features in our relatively simple photonic chip that has never been demonstrated before. This function and performance novelty is equally important as the device innovation or novel applications. **In our work**, the novelty lies in how these concepts and devices are used to unlock previously unachievable system performance.

2) The comparison with the electrical filters is not reasonable. Although NF and SFDR are not measured in Ref. [49] and Ref. [50], I believe they have better performances than the photonic filter, since they do not involve EO & OE conversions and active amplification.

Reply:

We respectfully disagree with this comment. First, we cannot make a fair comparison of the NF and SFDR of previous works, as they are not reported. Assuming that they have better performance would be speculative.

Recently, there are number of investigations about active RF bandpass filter with active-RC [1], active capacitance [2] or active inductor [3-5], that is built in CMOS technology. Nevertheless, the reported noise figure and dynamic range in these studies **are comparable** with our work (Table I).

This comparable result shows the uniqueness of programmable MWP circuit in our work. With simpler chip topology, **our work** can compete in term of performance, but with **better** wide tuning range, continuous tunability, and less hardware numbers that cannot be achieved easily by its electronic counterparts.

References:

- [1] B. Wu *et al.*, A 40 nm CMOS Derivative-Free IF Active-RC BPF With Programmable Bandwidth and Center Frequency Achieving Over 30 dBm IIP3, **IEEE Journal of Solid-State Circuits**, 50(8), 1772-1784, 2015.
- [2] D. Colaiuda *et al.*, A Second Order 1.8-1.9 GHz Tunable Active Band-Pass Filter with Improved Noise Performance, **Electronics**, 11(2781), 1-10, 2022.
- [3] Z. Gao *et al.*, A Fully Integrated CMOS Active Bandpass Filter for Multiband RF Front-Ends, **IEEE Transactions on Circuits and Systems II: Express Briefs**, 55(8), 718-722, 2008.
- [4] V. Kumar *et al.*, A 2.5 GHz Low Power, High-Q, Reliable Design of Active Bandpass Filter, **IEEE Transactions on Device and Materials Reliability**, 17(1), 229-244, 2017.
- [5] R. Mehra *et al.*, Reliable and Q-Enhanced Floating Active Inductors and Their Application in RF Bandpass Filter, **IEEE Access**, 6, 48181-48194, 2018.

The advantages of the proposed filter are the relatively higher gain and reconfigurability. However, the gain is realized by two EDFAs which will sacrifice the analog performance.

Reply:

We respectfully disagree with this comment. At this point, all MWP system is active and requires sufficient laser power and EDFA to compensate the optical losses in the system. The combination of EDFA with low-biasing Mach-Zehnder modulator (MZM) has been reported, can be used to improve the performance of the analog photonic system [1] which is **the opposite** of the reviewer comment. In fact, the positive result of having combination of EDFA and low-biasing MZM in the system is in-line with our work, as we reported the highest ever analog performance using our system.

Finally, the need for external EDFAs can be feasibly eliminated using high-power erbium-doped waveguide amplifiers (EDWA) in the silicon nitride (Si_3N_4) photonic integrated circuits, which is recently demonstrated by one of the co-authors of this work [2]. It can be reasonably foreseen in the future that applying the on-chip Erbium amplifier to the MWP chip will overcome the current necessities of the off-chip EDFAs, and at the same time it can enable demanded link gain and noise figure.

Reference:

- [1] V.J. Urick *et al.*, Analysis of an Analog Fiber-Optic Link Employing a Low-Biased Mach Zehnder Modulator Followed by an Erbium-Doped Fiber Amplifier, **IEEE Journal of Lightwave Technology**, 27(12), 2013-2019, 2009.

[2] Y. Liu *et al.*, A photonic integrated circuit-based erbium-doped amplifier, **Science**, 376, 1309-1313, 2022.

For the reconfigurability, I cannot clearly see (or find from the manuscript) the necessity of such a filter. Therefore, the significance of the work and the impact on the researchers in both MWP and microwave communities are not clear.

Reply:

Naturally, reconfigurable bandpass-bandstop filters are important for modern microwave and wireless systems with spectrally cognitive operation [1,2] and in modern multi-mode transceivers adaptable for several operation bands [3,4]. It is desirable to eliminate the need for large-volume switchable filter banks which are not unlimitedly scalable due to the limits in space, weight and power consumption. These filters can dynamically select RF signals of interest and mitigate frequency-agile interferers [5-7]. This need for reconfigurable filters has been clearly mentioned in the following supplementary references and has been motivating the progress of the topic of MWP. Additionally, the reconfigurable filter **in our work** can be achieved in **integrated form factor** as opposite with the usage of discrete components in references which gives our work the upper hand in term of the size and weight.

Recently, U.S. Defense Advance Research Project Agency (DARPA) announced a project called COFFEE or COmpact Front-end Filters at the EIEment-level to address the challenges hampering the use of wideband Active Electronically Scanned Array (AESA) in congested RF environments [8]. With this project, DARPA seeks to create new class of **integrable, high-frequency filters with low loss, high-power handling, and seamless uniformity** for superior electromagnetic spectrum operations in the modern era with one area of priority: **heightened multifunctionality and granular optimization** in AESAs [8]. The COFFEE project aims to develop integrable filters that operate over a wide range of frequencies, small enough to fit behind the element of phased array, and on the analog front-end to make the array in AESA more robust and resistant to interference before digital processing on the back-end [9].

References:

[1] E.J. Naglich *et al.*, A Tunable Bandpass-to-Bandstop Reconfigurable Filter with Independent Bandwidths and Tunable Response Shape, **IEEE Transactions on Microwave Theory and Techniques**, 58(12), 3770-3779, 2010.

[2] T. Yang *et al.*, Bandpass-to-Bandstop Reconfigurable Tunable Filters with Frequency and Bandwidth Controls, **IEEE Transactions on Microwave Theory and Techniques**, 65(7), 2288-2297, 2017.

[3] J. Lee *et al.*, New Bandstop Filter Circuit Topology and Its Application to Design of a Bandstop-to-Bandpass Switchable Filter, **IEEE Transactions on Microwave Theory and Techniques**, 61(3), 1114-1123, 2013.

[4] Y-H. Cho *et al.*, Two- and Four-Pole Tunable 0.7-1.1-GHz Bandpass-to-Bandstop Filters with Bandwidth Control, **IEEE Transactions on Microwave Theory and Techniques**, 62(3), 457-463, 2014.

[5] Y-C. Chiou *et al.*, A Tunable Three-Pole 1.5-2.2-GHz Bandpass Filter with Bandwidth and Transmission Zero Control, **IEEE Transactions on Microwave Theory and Techniques**, 59(11), 2872-2878, 2011.

[6] E.J. Naglich *et al.*, Switchless Tunable Bandstop-to-All-Pass Reconfigurable Filter, **IEEE Transactions on Microwave Theory and Techniques**, 60(5), 1258-1265, 2012.

[7] M. Fan *et al.*, Compact Bandpass-to-Bandstop Reconfigurable Filter with Wide Tuning Range, **IEEE Transactions on Microwave Theory and Techniques**, 29(3), 198-200, 2019.

[8] Defense Advance Research Project Agency. (2022), COFFEE Program Jump-Starts Integrable Filtering for Wideband Superiority, <https://www.darpa.mil/news-events/2022-06-01> , Last accessed on 2022 Sept.

[9] Defense Advance Research Project Agency. (2021), Filtering Out Interference for Next-Generation Wideband Arrays, <https://www.darpa.mil/news-events/2021-06-10> , Last accessed on 2022 Sept.

Revision:

We have clarified in the main text about the necessity of reconfigurable filter for modern RF applications as the following:

Introduction

As radio frequency (RF) and microwave systems are moving forward into cognitive operation, novel reconfigurable filter will become a key component to enable the full potential of these systems performance [1–3]. This filter can intelligently operate to differentiate RF signal of interest from the interferers [4–6]. There is a need of developing reconfigurable filter for modern RF systems to address the challenges impeding the use of active electronically scanned array (AESA) that operate at wide range of frequencies in dense RF environment [7]. The filter is aimed to make the array in AESA more resistant to interference before signal processing [8]. Integrated microwave photonic (MWP) can offer significant advantages to realize advanced concepts of reconfigurable filter for multi-band, all spectrum communications [9] and broadband programmable front-ends [10], which are important for modern RF communications (i.e. cognitive radio). To play a key role in modern RF applications, integrated MWP circuits need to simultaneously show advanced programmability and exceptional performance in terms of low losses, low noise figure, and high dynamic range in a reduced footprint [11–15]. In recent pasts, a number of programmable integrated MWP filters have widely been demonstrated [16–20]. Typically, these filters were achieved in application specific circuits, and the measured RF metrics were only sparsely reported. The values of the RF gain, noise figure (NF), and spurious-

free dynamic range (SFDR) in these circuits are usually far-off from the requirements for practical RF systems.

...

The capability of the proposed reconfigurable filter is expected to improve the system performance across S-band through Ku-band (i.e. 2 GHz to 18 GHz) frequency range and play a key role for the realization of practical programmable integrated MWP circuit that can operate in congested RF environment.

3) *From the Method, the power of the input two-tone RF signal is 8 dBm. According to Fig. 4(c), the power of the fundamental components are nearly -30 dBm, which cannot agree with the -26 dB claimed in Table I and the results shown in Fig. 4d. Please check it.*

Reply:

We thank the reviewer for the feedback about the two-tone signal. In the method section, the 8 dBm RF power is the output RF power from the signal generator. However, when we combine the RF signal from two signal generators, the RF combiner and the RF cable will introduce an 8.5 dB insertion loss, so the power sent to the phase modulator (the real input RF power) is around -0.5 dBm. The power of the fundamental components in Fig. 4(c) is -26.65 dBm. The difference between the output fundamental power and the real input RF power matches the link gain of the system, which is -26 dB.

We have updated the RF input power as -0.5 dBm in the method section.

For the SFDR measurements, we increased the input RF signal to observe the growing trends of the IMD3 and the fundamental terms after linearization. Then, we compare them with the changes of the IMD3 and the fundamental terms in the non-linearized system. The input RF power in Fig. 4(d) is the power sent to the phase modulator (PM). Here, we have already subtracted the insertion loss of the RF combiner and the RF cable.

Revision:

For the IMD3 suppression experiment, the two-tone RF signal with power of -0.5 dBm, centered at 9 GHz with a space of 10 MHz from signal generators (Wiltron 69147A and Rohde-Schwarz SMP02) is used to drive the PM.

4) *Although SFDRs at different frequencies are measured, it would be more convincing to directly input a wideband RF signal (for example, a QAM signal) to demonstrate its high dynamic range after filtering.*

Reply:

The presented SFDR is obtained from the standard measurement methods widely used in analog electronics RF/microwave [1-5] and analog microwave photonics [6-10], therefore, the measurement technique itself is reliable. We acknowledge the suggestion of the reviewer for wideband RF signal input. However, at this moment we do not have access to the experimental equipment capable of generating QAM signals and to measure their quality/error rate.

References:

- [1] C. Garcia-Alberdi *et al.*, Tunable Class AB CMOS $G_m - C$ Filter Based on Quasi-Floating Gate Techniques, **IEEE Transactions on Circuits and Systems I: Regular Papers**, 60(5), 1300-1309, 2013.
- [2] M. De Matteis *et al.*, A 33 MHz 70 dB-SNR Super-Source-Follower-Based Low-Pass Analog Filter, **IEEE Journal of Solid-State Circuits**, 50(7), 1516-1524, 2015.
- [3] J.S. Mincey *et al.*, Low-Power $G_m - C$ Filter Employing Current-reuse Differential Difference Amplifiers, **IEEE Transactions on Circuits and Systems II: Express Briefs**, 64(6), 635-639, 2017.
- [4] Y. Xu *et al.*, A 77-dB-DR 0.65-mW 20-MHz 5th-order Coupled Source Followers Based Low-Pass Filter, **IEEE Journal of Solid-State Circuits**, 55(10), 2810-2818, 2020.
- [5] M. De Matteis *et al.*, 64 dB Dynamic-Range 810 μ W 90 MHz Fully-Differential Flipped-Source-Follower Analog Filter in 28nm-CMOS, **IEEE Transactions on Circuits and Systems II: Express Briefs**, 68(9), 3068-3072, 2021.
- [6] G. Liu *et al.*, Integrated Microwave Photonic Spectral Shaping for Linearization and Spurious-Free Dynamic Range Enhancement, **IEEE Journal of Lightwave Technology**, 39(24), 7551-7562, 2021.
- [7] O. Daulay *et al.*, Microwave Photonic Notch Filter with Integrated Phase-to-Intensity Modulation Transformation and Optical Carrier Suppression, **Optics Letters**, 46(3), 488-491, 2021.
- [8] H. Shun Wen *et al.*, Ultrahigh Spectral Resolution Single Passband Microwave Photonic Filter, **Optics Express**, 29(18), 28725-28740, 2021.
- [9] R. Zheng *et al.*, Microwave Photonic Link with Improved Dynamic Range for Long-Haul Multi-Octave Applications, **IEEE Journal of Lightwave Technology**, 39(24), 7915-7924, 2021.
- [10] Y. Liu *et al.*, Tunable and Reconfigurable Microwave Photonic Bandpass Filter Based on Cascaded Silicon Microring Resonators, **IEEE Journal of Lightwave Technology**, 40(14), 4655-4662, 2022.

REVIEWERS' COMMENTS

Reviewer #2 (Remarks to the Author):

My comments have been reasonably addressed by the authors. The manuscript can be considered for publication.